# The Population-Wide Risk-Benefit Profile of Extending the Primary COVID-19 Vaccine Course Compared with an mRNA Booster Dose Program

**DOI:** 10.3390/vaccines10020140

**Published:** 2022-01-18

**Authors:** Tinevimbo Shiri, Marc Evans, Carla A. Talarico, Angharad R. Morgan, Maaz Mussad, Philip O. Buck, Phil McEwan, William David Strain

**Affiliations:** 1Health Economics and Outcomes Research Ltd., Cardiff CF23 8RB, UK; tinevimbo.shiri@heor.co.uk (T.S.); angharad.morgan@heor.co.uk (A.R.M.); phil.mcewan@heor.co.uk (P.M.); 2Diabetes Resource Centre, University Hospital Llandough, Cardiff CF64 2XX, UK; marclyndon1@hotmail.com; 3Moderna, Inc., Cambridge, MA 02139, USA; Carla.Talarico@modernatx.com (C.A.T.); Philip.Buck@modernatx.com (P.O.B.); 4Ashfield Healthcare on Behalf of Moderna, Ashby-de-la-Zouch LE65 1HW, UK; maaz.mussad@ashfieldhealthcare.com; 5Diabetes and Vascular Research Centre, University of Exeter Medical School, Exeter EX1 2HZ, UK; 6The Academic Department of Healthcare for Older Adults, Royal Devon and Exeter Hospital, Exeter EX2 5DW, UK

**Keywords:** boosters, modelling, COVID-19, population risk

## Abstract

The vaccination program is reducing the burden of COVID-19. However, recently, COVID-19 infections have been increasing across Europe, providing evidence that vaccine efficacy is waning. Consequently, booster doses are required to restore immunity levels. However, the relative risk–benefit ratio of boosters, compared to pursuing a primary course in the unvaccinated population, remains uncertain. In this study, a susceptible-exposed-infectious-recovered (SEIR) transmission model of SARS-CoV-2 was used to investigate the impact of COVID-19 vaccine waning on disease burden, the benefit of a booster vaccine program compared to targeting the unvaccinated population, and the population-wide risk–benefit profile of vaccination. Our data demonstrates that the rate of vaccine efficacy waning has a significant impact on COVID-19 hospitalisations with the greatest effect in populations with lower vaccination coverage. There was greater benefit associated with a booster vaccination strategy compared to targeting the unvaccinated population, once >50% of the population had received their primary vaccination course. The population benefits of vaccination (reduced hospitalisations, long-COVID and deaths) outweighed the risks of myocarditis/pericarditis by an order of magnitude. Vaccination is important in ending the COVID-19 pandemic sooner, and the reduction in hospitalisations, death and long-COVID associated with vaccination significantly outweigh any risks. Despite these obvious benefits some people are vaccine reluctant, and as such remain unvaccinated. However, when most of a population have been vaccinated, a focus on a booster vaccine strategy for this group is likely to offer greater value, than targeting the proportion of the population who choose to remain unvaccinated.

## 1. Introduction

Vaccination has played an important role in protecting against severe disease and death from COVID-19, as demonstrated in both clinical trials [1,2,3,4] and real-world settings [5,6,7,8,9]. Whilst vaccination has been highly effective at preventing severe disease, hospitalisation, and death due to COVID-19, it remains unclear how long immunity lasts following the primary course (two vaccine doses for the majority of the population, or three doses in immunocompromised individuals). Breakthrough infections are emerging in vaccine recipients [10,11,12] that are in part due to modest waning in protection against SARS-CoV-2 infection over time, as well as diminished antibody responses in those who are immunocompromised. This resulted in the re-instigation of lockdowns in several well-vaccinated European countries including Austria and Germany. In addition, breakthrough infections may be potentiated by an increase in infectivity of newer variants such as the Delta sub-lineage AY4.2, or the presence of multiple escape mutations as has occurred with the emergence of the Omicron variant [13,14]. A number of recent studies examining vaccine effectiveness and duration of protection against mild and severe COVID-19 report an increasing proportion of breakthrough infections among the earliest vaccinated individuals [15,16,17], with the greatest numerical reduction in vaccine efficacy in older adults and those in a clinical risk group [18].

The latest analysis from the ZOE COVID Study, which investigates real world vaccine effectiveness, examined data from positive PCR test results between May and July 2021 among 1.2 million people who had received two doses of the Pfizer/BioNTech or Oxford/AstraZeneca COVID-19 vaccine. These data demonstrated that protection following two doses of the Pfizer/BioNTech vaccine decreased from 88% at one month to 74% at five to six months (2.3–2.8% monthly decline assuming a linear reduction in efficacy), and following two doses of the Oxford/AstraZeneca vaccine decreased from 77% at one month to 67% at four to five months (2.0–2.5% monthly decline assuming a linear reduction in efficacy) [19,20]. The Moderna mRNA-1273 vaccine similarly demonstrates waning efficacy, albeit not as pronounced, from 94.1% at 14–60 days after vaccination to 80.0% at 151–180 days [21]. These findings suggest that booster doses are required to restore the initial high amounts of protection observed early in the vaccination programme.

Many countries have already started rolling out COVID-19 vaccine boosters, with others expected to soon follow suit. The U.S. Food and Drug Administration (FDA) has authorised both the Moderna and Pfizer-BioNTech COVID-19 vaccines as a single booster dose for all individuals ≥18 years, after completion of primary vaccination with any FDA-authorised or approved COVID-19 vaccine [22], and consequently the Centers for Disease Control and Prevention’s (CDC) Advisory Committee on Immunization Practices (ACIP) has expanded recommendations for booster vaccines for people ≥18 years [23]. The Joint Committee on Vaccination and Immunisation (JCVI) in the UK has provided advice on a COVID-19 vaccine booster programme for frontline health and social care workers, adults over 40, and younger adults with underlying health conditions, who had their second COVID-19 vaccine at least six months earlier [24,25,26]. The advice is based on early data from the COV-BOOST study, which studied the use of seven different COVID-19 vaccines when given as a third booster dose to 2883 participants considered representative of the UK population who have received their primary COVID-19 vaccinations [27]. Initial data indicated that booster doses of COVID-19 vaccines are generally well tolerated and provide a substantial increase in vaccine-induced immune response [28]. A larger Israeli study on the efficacy of third vaccine doses extracted data from the Israeli Ministry of Health database (30 July to 31 August 2021) for more than 1.1 million people and compared the incidence of confirmed COVID-19 and severe illness in those who had received the booster dose at least 12 days earlier with the rates in people who had not received a booster. The rate of confirmed infection and severe illness was lower in the booster group than in the non-booster group by a factor of 11.3 (95% confidence interval 10.4 to 12.3) and 19.5 (12.9 to 29.5), respectively [29].

COVID-19 boosters can cause similar adverse reactions to those observed with previous doses [30]. For the most part, compared to COVID-19 infection and the risk of long COVID and other severe outcomes, these vaccine side effects are likely to be mild and temporary (pain at the injection site, fatigue, muscle pain, headache, and fever) and are unlikely to cause hospitalisation. More serious adverse events including myocarditis associated with both the Pfizer and Moderna mRNA COVID-19 vaccines have been observed, predominantly in young males after the second dose [31]. Vaccine induced thrombosis with thrombocytopenia syndrome (VITTS) and Guillain-Barre syndrome (GBS) have been associated with adenovirus-vectored COVID-19 vaccines [32,33], although concern regarding the possibility of immunity to the vector is limiting the use of these in the booster program. Although the risk of these adverse events was significantly less than the risks of COVID-19 infection itself in the initial waves, there is uncertainty regarding their impact in a population that has some residual protection from their initial vaccination course.

Whilst the risk–benefit ratio of COVID-19 vaccination has been evaluated at the individual level, the risk–benefit ratio of primary series vaccinations and boosters have not yet been quantified for populations. As a result, in response to rising COVID-19 infection rates, international strategies are divided between focusing on a booster vaccination programme and imposing population mixing restrictions. With this in mind, the objective of this study is to evaluate the impact of COVID-19 vaccine efficacy waning on disease burden, and subsequently quantify the population-wide risk–benefit profile of COVID-19 vaccination (primary series and boosters). In addition, we aimed to determine the relative impact on COVID-19 hospitalisation rates of a vaccine strategy focused on providing boosters to those who have previously received the primary course more than six months earlier, compared to attempting to reach the unvaccinated population with a primary course. Given the integrated primary care record system of the UK, with over 98% of the population registered we used the UK primary care records to evaluate the impact of either strategy in a global system.

## 2. Methods

### 2.1. Disease Transmission Model Overview

Our previously published Susceptible-Exposed-Infectious-Recovered (SEIR) transmission model of SARS-CoV-2 virus [34] was updated to capture the impact of immunity waning and booster dosing of those who had a primary series of COVID-19 vaccination (Figure 1). The model was populated with UK data as a reference case. However, the model includes important and generalisable dynamics and as such the results are likely to be applicable globally. In addition to the epidemiological parameters (population distribution by age, mixing patterns, restriction information, vaccine coverage and speed of vaccine rollout, immunity waning), model inputs included current infection levels in the population, achievable vaccine coverage forecast and rollout speed in the population. Further details regarding the disease transmission model, including the input parameters and assumptions, can be found in the supplementary material (Appendix A).

### 2.2. Investigating the Impact of COVID-19 Vaccine Waning on Disease Burden

To illustrate the impact of immunity waning on disease burden in the UK in the next three months, the following variables were examined: incidence of infection between 100–600 cases per 100,000 population, current vaccine coverage in the population from 20 to 80%, linear vaccine immunity waning against transmission from 0 to 12% per month.

The vaccine rollout speed was assumed to be 0.3% per day since the start of the UK vaccination programme [35], up to current coverage levels, and was assumed to be 0.1% per day for the next three months. The maximum achievable coverage in the population in the next three months was assumed to be 90% across all age groups. Vaccine efficacy against transmission and severe cases (hospitalisations and deaths) was assumed to be 88% and 91%, respectively (without waning).

### 2.3. Quantifying the Risk–Benefit Profile of COVID-19 Vaccination at the Population Level

To quantify the population-wide risk–benefit profile of COVID-19 vaccination (primary series and boosters) a decision tree model was employed (Appendix A). In the model, vaccinated and unvaccinated individuals may remain un-infected or may develop mild or severe COVID-19 infection. The most potentially serious vaccine-associated adverse event from either the Pfizer or Moderna mRNA vaccines was considered to be myocarditis/pericarditis. Age and gender specific risk of this occurring was derived from the CDC database (crude reporting rates to the Vaccine Adverse Event Reporting System (VAERS) following mRNA COVID-19 vaccination) (Appendix A). An adverse event rate is constant whereas severe outcomes due to COVID-19 are time dependent as they depend on vaccine parameters and infection rates. The impact of these factors on the threshold for the vaccine being detrimental (adverse events outweighing vaccine benefits) was assessed.

Using the dynamic model, the following assumptions were applied: primary series coverage between 10 and 80% on all age groups except 0–9 years age (not vaccinated), 90% maximum achievable coverage in the population in three months, incidence of infection from 10 to 600 per 100,000 population (current figure in the UK 523/100,000 cases), vaccine efficacy against severe cases of 91% (pre-modelling for waning) and vaccine efficacy against infection of 88%. Double vaccinated individuals were modelled to reduce onward transmission by 52% against the dominant Delta variant. Waning efficacy was considered to be symmetrical across all parameters, including hospitalisation, mortality, and onward transmission.

## 3. Results

### 3.1. Impact of Vaccine Waning on COVID-19 Hospitalisations

Assuming no waning of vaccine efficacy in the model and assessing the expected number of COVID-19 hospitalisations within three months as a function of baseline infection rates and coverage, we observed that increasing infection rates are associated with increasing hospitalisations (Appendix A). Unsurprisingly, the majority of these hospitalisations occurred in the unvaccinated populations and were reduced by increasing vaccination coverage.

In Figure 2, the additional hospitalisations driven by different modeled waning of vaccination efficacy on transmission risk in the population is demonstrated. The rate of waning of vaccination efficacy has a significant impact on hospitalisations with the effect largest in populations with lower vaccination coverage. For populations with high coverage there is a minimal effect on hospitalisations even with significant waning.

### 3.2. Balancing the Benefit of Targeting Primary Series or Booster Vaccine Program

We evaluated the relative benefits of targeting a booster approach compared to attempting to reach the unvaccinated population with a primary course on hospitalisation rates. This factored the different vulnerability of those that received their primary course more than six months ago, based on the JCVI hierarchy of need and timetable of roll-out (Figure 3). The results demonstrated that, once ~50% of the population had received their primary vaccination course there was greater benefit in terms of reducing hospitalisation, by offering those that had already received their primary course a booster, compared to targeting the unvaccinated population. As vaccine waning increased the threshold benefit of boosters was reduced (Appendix A; Appendix A). For example, if the efficacy of primary vaccine was waning by 12% per month or more, the benefits of the booster approach were apparent when ~40% of the population had received their primary vaccination course.

*Results are based on current estimates of vaccine efficacy waning of 2.5% per month [19,20,21]*.

### 3.3. Population-Wide Risk–Benefit Profile of COVID-19 Vaccination

The risk of severe adverse events (myocarditis/pericarditis) following primary series and booster vaccination over a three-month vaccine campaign was compared with the cumulative number of COVID-19 hospitalisations, long COVID and deaths during the same period (Figure 4). Although at an individual level, the immediate risk of myocarditis/pericarditis appears significant, both the population and personal risk associated with severe disease (hospitalisations, long COVID and deaths) remains considerably higher when SARS-CoV-2 rates are above 10 cases per 100,000 population. Our data demonstrate that myocarditis/pericarditis is more likely in the 10–19 and 20–29 year-old age groups, albeit at low rates. This mirrors the current understanding about the incidence of myocarditis following COVID-19 mRNA vaccines. When community rates of SARS-CoV-2 are very low (≤10 cases per 100,000 population), the risks associated with vaccination become greater than the benefit. To put this into perspective, as of 23rd November 2021, current community prevalence is approximately 420/100,000 in the UK [36], 390/100,000 in Germany [37], and 860/100,000 in Belgium [37]. At these infection levels, the personal and population benefits of vaccination far outweigh the risks of adverse effects by an order of magnitude for all age groups.

## 4. Discussion

Our unique, population-based model has demonstrated the potential impact of different COVID-19 vaccination strategies dependent on the community incidence of SARS-CoV-2, the vaccination rate within the local population and the degree of waning of vaccine efficacy. The observation that lower penetration of vaccination is associated with higher hospitalisations in both the unvaccinated and vaccinated population is expected. Further, the reduction in hospitalisation and mortality by more than 50% for every 20% increase in completed courses of vaccination represents the wider benefit of reducing the viral reservoir in the community and the modest reduction in asymptomatic transmission afforded by vaccination. The results of our study also demonstrated that the impact of waning vaccine efficacy is amplified in populations with poorer vaccination uptake. As such, once the vaccination rate in the population has achieved 50%, there is greater societal benefit from pursuing a booster strategy for the vulnerable individuals that have already received a complete vaccination course rather than further pursuit of the primary course. Clearly this is based on a population risk rather than personal risk, as there is raw data from the UK Health Security Agency demonstrating that those vaccinated have a lower risk of admission to hospital than their unvaccinated counterparts [38]. Furthermore, in all scenarios studied, the benefits of vaccination outweighed the risks of serious adverse effects, i.e., myocarditis/pericarditis, by an order of magnitude, indeed community prevalence had to be reduced to near elimination levels of <10/100,000 population before the risk of vaccination reaches equipoise with the risk of adverse effects from COVID-19.

No country or region has achieved 100% COVID-19 vaccination rates, despite public health messaging and appropriate incentivisation. The reasons for failure to attain complete vaccine coverage are complex and multifactorial. With other vaccination programs, such as influenza, proposed explanations have included a perceived lack of salience, lack of access, or belief in vaccine myths such as the disease not being a serious health concern [39]. Given the global impact of the COVID-19 pandemic, and the availability of free vaccines in the vast majority of healthcare systems, it is unlikely that these reasons account for the proportion of the population who remain unvaccinated against COVID-19. Instead, it is more likely that those who remain unvaccinated are either convinced by some of the misinformation that vaccination with these new technologies carries more risk than benefit (“anti-vax”), or uncertain as to the benefit and how it impacts them directly (“vaccine hesitant”). There are substantial differences in the approach to addressing these individuals. Whereas the latter can often be persuaded to receive treatment with reason and evidence, the former requires considerable more time, and, even then, are unlikely to believe the protection vaccination offers. Conversely, people who have already come forward for their primary course are likely to self-present for their booster dose. As such, there is an incremental time commitment involved in reaching different groups, with those that have already received a full course taking only a small investment of health care professional resources, the vaccine hesitant requiring more time and the anti-vax population considerably more, and even then, often without success.

As we enter a global “post-pandemic” recovery phase with health care resources refocussing in the management of long-term conditions, such as hypertension, diabetes and heart failure, resources to establish a comprehensive vaccine program have become limited, or may remain limited in resource-poor settings. As a result, health care providers and policy makers are currently reviewing where their focus should be in the subsequent distribution of COVID-19 vaccinations. Data from our model suggest that at lower vaccination rates in the community, the effect of high transmission is likely to have an adverse effect on the entire community and the health system. Therefore, in this scenario it would be appropriate to focus limited resources on achieving greater primary series vaccine penetration, rather than promoting booster vaccines, which although offering up to an additional 95% protection to individuals would have very little impact on the overall health utilisation burden. However, once primary series vaccine penetration achieves a critical level of ~50%, the investment in booster vaccination has a greater impact on hospitalisations, and therefore the wider health system, than trying to achieve incremental gains in coverage. Thus, in countries or regions where more than 50% of the population has received their primary course, whilst completing a primary course may have a greater impact on a personal level for the recipients, vaccinating a greater number of individuals who present for their booster is likely to offer greater value to healthcare systems.

The key side effect of vaccination with the mRNA-based technologies is myocarditis, particularly in males <40 years old and after the second dose. This cardiac inflammation is a well-recognised complication of COVID-19 infection itself, thought to be due to the inflammatory response associated with the interplay between the spike protein, trans-membrane protein serinase-2 and the angiotensin-converting enzyme 2. As a result, it is unsurprising that a milder form of this complication occurs with a vaccine that promotes endogenous production of the same spike protein. During the first 12 months of the pandemic, 12 to 17 year-old males were most likely to develop myocarditis within three months of COVID-19 infection, at a rate of about 450 cases per million infections compared with 77 cases of myocarditis per million males of the same age following their first or second dose of a Pfizer/BioNTech or Moderna vaccine, a sixth of that seen after infection [40]. Throughout our analysis, myocarditis/pericarditis did not, at any point, come close to the risks associated with contracting COVID-19 (hospitalisations, long COVID and deaths).

Our study has several limitations. Firstly, our modelling study is based on a hierarchical distribution of vaccine, with those at highest risk of an adverse outcome and hospitalisation from COVID-19 infection vaccinated first. The benefits observed in the data may not be as great in healthcare systems where there was no prioritisation for vaccination of people at greatest risk of hospitalisation. Secondly, we also did not consider the differential time resource that it would take to achieve vaccination in the vaccine hesitant, or those with anti-vaccination sentiments. For these individuals, it is likely that the benefit from a targeted booster vaccination program will be greater. Finally, we only considered the direct impact of contracting SARS-CoV-2 on hospitalisation due to COVID-19, and did not consider any secondary impacts of infection, independent of primary disease severity, e.g., secondary diabetes, or the multiple thromboembolic conditions that have been reported. Given that there are such diverse reports as to the impact of vaccination on these conditions, it is difficult to measure the impact. However, as the principle focus of the majority of health economies is currently protecting their hospital services and there remains significant uncertainty regarding the severity and longevity of these secondary outcomes of SARS-CoV-2 infection, and thus health economic impact, this is not currently part of the UK, nor many other nation’s strategic considerations.

In conclusion, we have developed a mathematical model comparing the relative merit of a vaccination strategy based on targeting unvaccinated individuals to receive their primary vaccine series versus offering boosters to the already vaccinated population who are likely experiencing waning vaccine efficacy. Data from the model suggest that, providing the community vaccination penetration is at least 50%, there is greater benefit to the population at large in a booster vaccine strategy than attempting to reach those who been reluctant, or resistant to come forward for vaccination. The population-wide benefits of both primary series and booster vaccinations significantly outweigh any risks; this critical information can be used by countries and regions globally to determine prospective vaccination policy.

## Figures and Tables

**Figure 1 vaccines-10-00140-f001:**
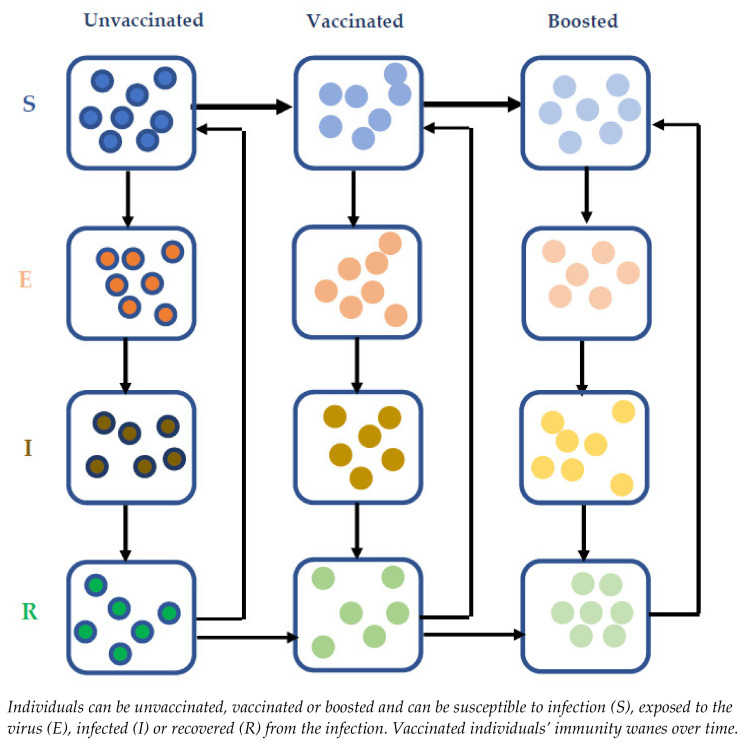
Model scheme for vaccinating framework.

**Figure 2 vaccines-10-00140-f002:**
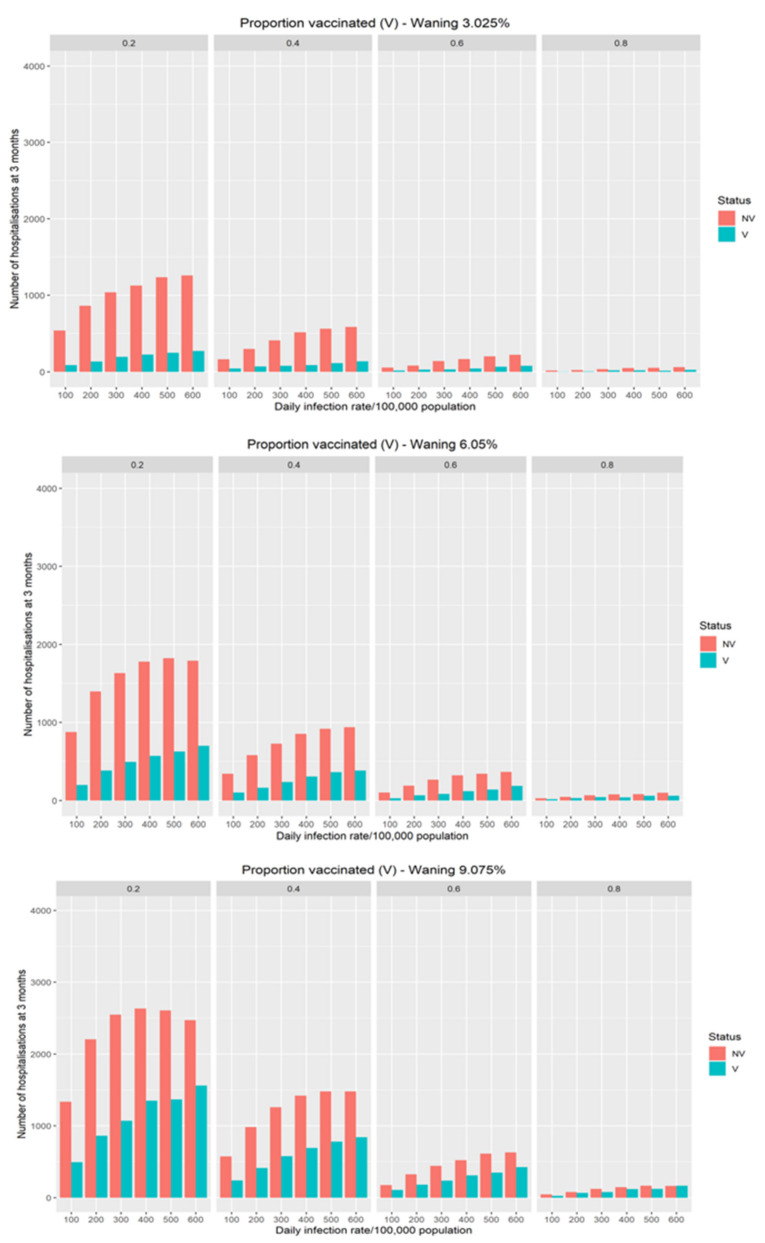
The impact of different theoretical waning rate on the expected number of daily COVID-19 hospitalisations at three months as a function of infection rates and vaccine coverage.

**Figure 3 vaccines-10-00140-f003:**
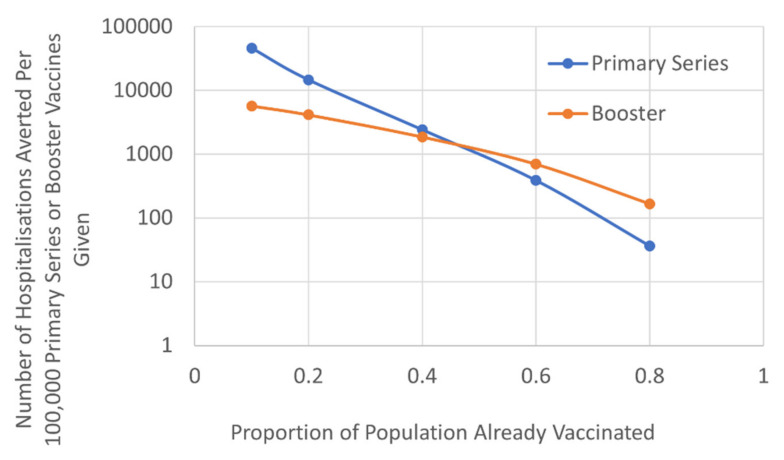
The number of COVID-19 hospitalisations prevented per 100,000 primary series or booster vaccines (logarithmic scale).

**Figure 4 vaccines-10-00140-f004:**
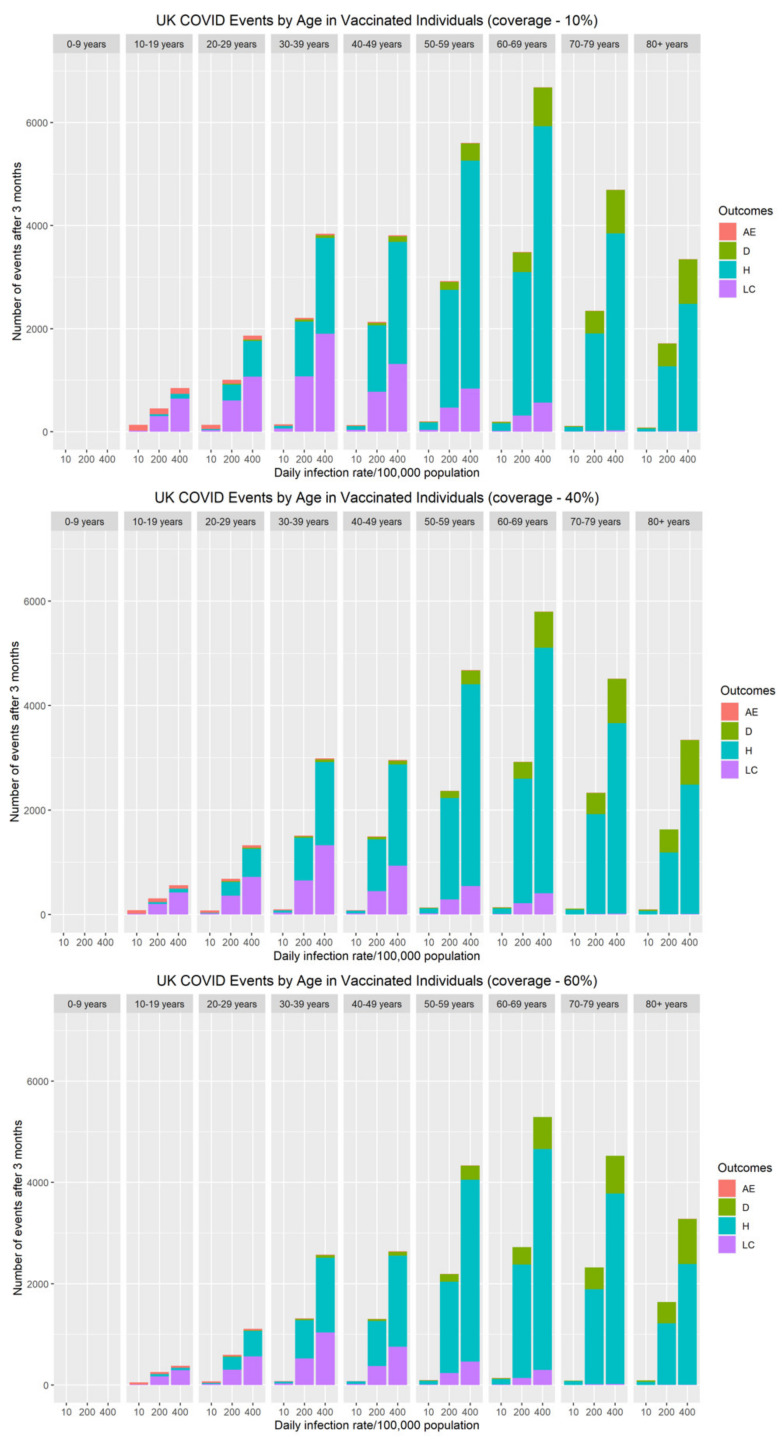
The impact of vaccine coverage and infection rates on the total number of adverse events (myocarditis/pericarditis), hospitalisations, long COVID and deaths after three months by age in vaccinated individuals in the population.

## Data Availability

The datasets analysed during the current study were sourced from and are available in the original publications referenced.

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
