# Peer review of "The Population-Wide Risk-Benefit Profile of Extending the Primary COVID-19 Vaccine Course Compared with an mRNA Booster Dose Program"

_vaccines, 2022, doi:10.3390/vaccines10020140_

Round 1

Reviewer 1 Report

Several countries have already started booster vaccination against COVID-19. However, a final decision on this has not been reached by all countries. The authors have developed a model to evaluate risk-benefit of two approaches for controlling CIVD-19 infections; booster vaccination with mRNA vaccine and imposing population mixing restrictions. Using the developed models, the authors have assessed the benefits of booster vaccination with mRNA vaccines for different vaccinated groups. Also, the authors have discussed target population for booster vaccination or primary course (vaccination of unvaccinated people). The results have implications for making an informed decision regarding future rounds of COVID-19 vaccination.

  1. Lines 44-52. The authors should also explain that the breakthrough infections could be due to the emergence of scape mutants of the virus not just waning the protection. The current vaccines contain original variants (inactivated whole virus, isolated at the beginning of the pandemic and so new mutants may scape the protection obtained from current vaccines.
  2. Line 93 and 168. As done when discussing the efficacy of mRNA vaccines, it is appropriate to specify mRNA vaccines (for example, Moderna and Pfizer mRNA vaccines) when discussing their side effects.
  3. Line 223. Do the authors mean 20-29?

Author Response

We thank the reviewers for their comments. We have addressed these point by point below. We agree that these amendments improve the overall manuscript

Reviewer 1

Several countries have already started booster vaccination against COVID-19. However, a final decision on this has not been reached by all countries. The authors have developed a model to evaluate risk-benefit of two approaches for controlling CIVD-19 infections; booster vaccination with mRNA vaccine and imposing population mixing restrictions. Using the developed models, the authors have assessed the benefits of booster vaccination with mRNA vaccines for different vaccinated groups. Also, the authors have discussed target population for booster vaccination or primary course (vaccination of unvaccinated people). The results have implications for making an informed decision regarding future rounds of COVID-19 vaccination.

  1. Lines 44-52. The authors should also explain that the breakthrough infections could be due to the emergence of scape mutants of the virus not just waning the protection. The current vaccines contain original variants (inactivated whole virus, isolated at the beginning of the pandemic and so new mutants may scape the protection obtained from current vaccines.

We thank you for this comment. As you can probably imagine, this manuscript was originally written before the emergence of Omicron variant, at a time when the vaccine had demonstrated efficacy against all of the existing variants. We have amended the manuscript accordingly including appropriate referencing

  1. Line 93 and 168. As done when discussing the efficacy of mRNA vaccines, it is appropriate to specify mRNA vaccines (for example, Moderna and Pfizer mRNA vaccines) when discussing their side effects.

A very reasonable comment. In both of the associated references the rate of myocarditis and pericarditis is not differentiated between the Pfizer and Moderna mRNA vaccine, therefore we have specified this.

  1. Line 223. Do the authors mean 20-29?

We apologise for this transcription error. You are correct this is the 20-29 year olds and has been appropriately adjusted.

Reviewer 2 Report

This paper shows that there was a greater benefit associated with a booster vaccination strategy compared to targeting the unvaccinated population, once >50% of the population had received their primary vaccination course. Therefore, the authors claim that a focus on a booster vaccine strategy for vaccinated people is likely to offer greater value, than targeting the proportion of the population who choose to remain unvaccinated.

General comment

The Paper is well written and ideas and concepts are adequately presented through the text. However, it is not really clear in the text the interpretation that “a focus on a booster vaccine strategy for this group is likely to offer greater value, than targeting the proportion of the population who choose to remain unvaccinated”. Authors should present the results in a more categoric way to show solid interpretations.

Specific points

It is not clear through the text how the authors estimated the waning rate (%) shown in Fig. 2.

Data from ZOE study showing that protection following two doses of the Pfizer/BioNTech vaccine decreased from 88% at one month to 74% at five to six months, and following two doses of the Oxford/AstraZeneca vaccine decreased from 77% at one month to 67% at four to five months is only presented as press released and not publish in a peer-reviewed scientific journal. In fact, it is known that vaccines does not prevent the infection, but reduces the progression of the disease and hospitalization, consistent with the findings that vaccination with an mRNA COVID-19 vaccine was significantly less likely among patients with COVID-19 hospitalization and with disease progression (https://jamanetwork.com/journals/jama/fullarticle/2786039 ).

Therefore, when the authors claim in the Introduction section that “These findings suggest that booster doses are required to restore the initial high amounts of protection observed early in the vaccination programme” may be removed since data available in the literature (peer-reviewed) do NOT support these findings.

Reference 17 and 21 should NOT be included since they are pre-prints and have not been peer-reviewed.

Reference 18 does not support the fact that the authors claim the “greatest reduction in vaccine efficacy in older adults and those in a clinical risk group”.

Author Response

We thank the reviewers for their comments. We have addressed these point by point below. We agree that these amendments improve the overall manuscript

Reviewer 2

This paper shows that there was a greater benefit associated with a booster vaccination strategy compared to targeting the unvaccinated population, once >50% of the population had received their primary vaccination course. Therefore, the authors claim that a focus on a booster vaccine strategy for vaccinated people is likely to offer greater value, than targeting the proportion of the population who choose to remain unvaccinated.

 General comment

The Paper is well written and ideas and concepts are adequately presented through the text. However, it is not really clear in the text the interpretation that “a focus on a booster vaccine strategy for this group is likely to offer greater value, than targeting the proportion of the population who choose to remain unvaccinated”. Authors should present the results in a more categoric way to show solid interpretations.

Thank you for this assessment. We have now added a sentence in the first paragraph of the discussion that explicitly states this.

Specific points

It is not clear through the text how the authors estimated the waning rate (%) shown in Fig. 2.

We apologise for this. The percentage waning efficacy demonstrated in figure 2 are the arbitrary percentage wanings that we estimated in our models. They should be regarded as a sensitivity analyses for different hypothetical waning efficacy levels. The model can then be adjusted for any level of waning, particularly as new variants arise that have different efficacy and waning characteristics. We have clarified this in the text and in the legend of the figure

Data from ZOE study showing that protection following two doses of the Pfizer/BioNTech vaccine decreased from 88% at one month to 74% at five to six months, and following two doses of the Oxford/AstraZeneca vaccine decreased from 77% at one month to 67% at four to five months is only presented as press released and not publish in a peer-reviewed scientific journal. In fact, it is known that vaccines does not prevent the infection, but reduces the progression of the disease and hospitalization, consistent with the findings that vaccination with an mRNA COVID-19 vaccine was significantly less likely among patients with COVID-19 hospitalization and with disease progression (https://jamanetwork.com/journals/jama/fullarticle/2786039).

Therefore, when the authors claim in the Introduction section that “These findings suggest that booster doses are required to restore the initial high amounts of protection observed early in the vaccination programme” may be removed since data available in the literature (peer-reviewed) do NOT support these findings.

We have modified the text slightly based on the peer reviewed and published analysis in Lancet ID based on the original press release.

Antonelli M, Penfold RS, Merino J, Sudre CH, Molteni E, Berry S, Canas LS, Graham MS, Klaser K, Modat M, Murray B, Kerfoot E, Chen L, Deng J, Österdahl MF, Cheetham NJ, Drew DA, Nguyen LH, Pujol JC, Hu C, Selvachandran S, Polidori L, May A, Wolf J, Chan AT, Hammers A, Duncan EL, Spector TD, Ourselin S, Steves CJ. Risk factors and disease profile of post-vaccination SARS-CoV-2 infection in UK users of the COVID Symptom Study app: a prospective, community-based, nested, case-control study. Lancet Infect Dis. 2022 Jan;22(1):43-55. doi: 10.1016/S1473-3099(21)00460-6. Epub 2021 Sep 1. PMID: 34480857; PMCID: PMC8409907.

Reference 17 and 21 should NOT be included since they are pre-prints and have not been peer-reviewed.

Reference 17 has been deleted, as the point is supported by reference 18-20. Reference 21 has since been peer reviewed, accepted and published in the BMJ and has been modified in the reference list accordingly

Reference 18 does not support the fact that the authors claim the “greatest reduction in vaccine efficacy in older adults and those in a clinical risk group”.

This reference has been updated with Collier, D.A., Ferreira, I.A.T.M., Kotagiri, P. et al. Age-related immune response heterogeneity to SARS-CoV-2 vaccine BNT162b2. Nature 596, 417–422 (2021).